# Unitary Space-Time Modulation Based on Coherent Codes

**DOI:** 10.3390/s22239049

**Published:** 2022-11-22

**Authors:** Daniel Calabuig

**Affiliations:** Instituto de Telecomunicaciones y Aplicaciones Multimedia (iTEAM), Universitat Politècnica de València, 46022 Valencia, Spain; dacaso@iteam.upv.es

**Keywords:** constellation design, noncoherent multiple input–multiple-output (MIMO) communication, pilot symbol-assisted modulation

## Abstract

This paper analyzes the relationship between pilot symbol-assisted modulation (PSAM) and unitary space-time modulation (USTM). In particular, we present a map that transforms any PSAM into a USTM and vice versa. USTMs are known to be capacity-achieving. However, most of the proposed USTM construction methods in the literature are computationally expensive, and the resulting constellations do not have a known structure that could simplify their decoding. Using the relationship between PSAM and USTM, and inspired by a graphical representation of these constellations used in this paper, we propose new USTM construction methods, which ensure that the USTM has a good performance compared to the corresponding PSAM, and a feasible construction and decoding, even for high data rates.

## 1. Introduction

Most of the current communication systems are based on coherent reception where channel state information (CSI) should be estimated for equalization, demodulation, etc. The CSI can be obtained via training by inserting pilots in the data signals. These schemes, however, increase the signaling overhead, which can be counterproductive in systems with many antennas or fast movement of either side of the communication link [1].

This drawback increased the interest in schemes that do not require full CSI at the transmitter and the receiver. An information-theoretic analysis of multi-antenna Rayleigh block-fading channels was performed in [2]. Motivated by this work, a communication method using unitary space-time modulation (USTM) and noncoherent reception was proposed in [3]. In this method, assuming that the receiver has *N* antennas, the transmitter has M≤N antennas, and the channel block length is T≥M+N channel uses, the transmitted signals, viewed as matrices with T×M elements, form a unitary matrix, i.e., one with orthonormal columns. This signal structure was shown to be capacity-achieving for high signal-to-noise ratio (SNR) or T≫M [3]. USTM constellations can be constructed minimizing the pairwise error probability of the constellation elements, although this is, in general, a difficult task. Other construction methods were inspired by the geometrical interpretation, given in [4], of the capacity expression, i.e., sphere packing in the Grassmann manifold G(T,M): the set of all *M*-dimensional subspaces of $C^{T}$. In particular, the columns of the USTM matrices are viewed as a basis that spans an *M*-dimensional subspace. At the receiver, the channel modifies the basis but keeps the subspace unchanged. Therefore, information should be encoded into subspaces and not into the particular basis. This idea is used in [5] to define a superposition coding of several USTM constellations in Grassmann manifolds.

These results motivated the USTM contellation design as sphere packing in the Grassmann manifold by using either numerical optimization tools [3,6,7,8,9] or algebraic constructions [10,11,12]. The main advantage of the first approach is that it does not restrict the constellation to have a specific structure. However, the direct optimization is computationally expensive, and the lack of structure implies that all the USTM matrices have to be stored and that the optimum receiver has to test all signals. The second approach equips the constellations with a certain structure, although with either poor performance [10], intractable decoding [11], or intractable constellation construction for a high number of degrees of freedom [12]. A different approach to constructing USTM constellations is to map coherent codes, i.e., those designed for coherent reception, into the Grassmann manifold. An example in which the exponential map is used can be found in [13]. This map has the drawback of not being one-to-one, requiring an appropriate scaling of the coherent codes which is, in general, not straightforward. Another example can be found in [14]. In this example, the coherent codes are mapped in the faces of a hypercube, which are projected into a hypersphere. Although the final USTM has a good performance, this technique can only be used with one transmit antenna. A similar approach was proposed in [15]. In this case, the authors use a different hypercube, but the technique can only be used with, again, one transmit antenna.

Due to the inability of producing sphere packings in the Grassmann manifold with efficient decoding and good performance, various works proposed the combination of training pilots and coherent codes, known as pilot symbol-assisted modulation (PSAM), as a meaningful alternative [4,16,17,18,19]. However, in this case, the resulting matrices are no longer unitary, and hence, the constellations are not capacity-achieving, although, in some cases, they can be shown to be similar to other USTMs [20]. This fact suggests that USTMs and PSAMs might not be as different as originally thought. In fact, the USTM constellation in [21] was shown to be more easily interpreted and analyzed as a PSAM [20].

In this paper, we use a map that transforms any PSAM into a USTM and vice versa, and hence generalize the observation done in [20] for one particular case. To obtain this map, we propose a new definition of the space-time matrices that compose a PSAM. Moreover, this definition enables a graphical representation of both USTMs and PSAMs that will draw insight into their relationship. Inspired by this representation and the PSAM matrix definition, we propose a new USTM constellation construction method that outperforms and inherits the structure of a coherent code. This construction method ensures that the USTM has a good performance compared to the coherent code, and a feasible construction and decoding, even for high data rates.

The rest of the paper is organized as follows. In Section 2, we present the system model and introduce both the graphical representation and the new definition of PSAM matrices. In Section 3, we present the map to transform PSAMs into USTMs and vice versa. In Section 4, we present a first USTM construction method and use the graphical representation to show the limitations of this method. In order to overcome these limitations, we propose a second USTM construction method in Section 5. In Section 6, we compare the performance of particular constellations constructed with the previous methods. Finally, main conclusions are drawn in Section 7.

## 2. System Model and Constellation Structure

Consider a wireless communication system with *M* transmit and *N* receive antennas, and a block-fading channel with a coherence interval of *T* channel uses. Let $X\in C^{T\times M}$ and $Y\in C^{T\times N}$ be the transmitted and received signals during *T* constant-fading channel uses, respectively, where the transmitted signal satisfies the power constraint
(1)$$E\left[{\left||X|\right|}_{F}^{2}\right]=M,$$
and ${\left||X|\right|}_{F}=\sqrt{\mathrm{tr}(X^{*}X)}$ is the Frobenius norm. The relationship between X and Y is given by
(2)$$Y=XH+\sqrt{\frac{M}{\rho T}}Z,$$
where $H\in C^{M\times N}$ is the channel matrix whose entries are i.i.d. drawn from the standard complex Gaussian distribution CN(0,1), $Z\in C^{T\times N}$ is the additive noise whose entries are also i.i.d. drawn from CN(0,1), and ρ is the signal-to-noise ratio (SNR).

The transmitted signals are selected from a constellation of matrices with $L=2^{RT}$ elements, where *R* is the transmission rate. The constellation $C^{U}={\left\{X_{i}^{U}\right\}}_{i=1}^{L}$ is a USTM if the elements are T×M unitary matrices, i.e., $X_{i}^{U*}X_{i}^{U}=I_{M}$, where $I_{M}$ is the M×M identity matrix, and they span different subspaces, i.e., $S\left(X_{i}^{U}\right)\neq S\left(X_{j}^{U}\right)$ for all i≠j, where $S\left(X\right)=\left\{Xx,x\in C^{M}\right\}$ is the subspace spanned by the columns of X. In the case of a PSAM, a certain quantity of channel uses are reserved to carry the pilot signals. If optimization over the training and data powers is allowed, the optimal number of training channel uses is *M* [1]. In this case, T−M channel uses are used to transmit matrices from a coherent code $A={\left\{A_{i}\right\}}_{i=1}^{L}$, where $A_{i}\in C^{T-M\times M}$. Therefore, the PSAM can be constructed concatenating the training and data signals in A. However, it is also possible to allow these signals to spread throughout all the coherence intervals by means of a unitary matrix. In particular, let $\left[B_{1}\;B_{2}\right]$ be a T×T unitary matrix, where $B_{1}$ and $B_{2}$ are the matrices with the first *M* columns and the last T−M columns, respectively. The constellation $C^{P}={\left\{X_{i}^{P}\right\}}_{i=1}^{L}$, where
(3)$$X_{i}^{P}=B_{1}P+B_{2}A_{i},\;i=1,\ldots ,L,$$
and $P\in C^{M\times M}$ is a pilot matrix, is a PSAM. In order to maximize the achievable rates, P should be a scaled unitary matrix [1]. Note that, in order to satisfy the power constraint in (Equation 1), assuming that all the matrices in the constellation are equiprobable, the coherent code and the pilot matrix must satisfy $\frac{1}{L}\sum_{i=1}^{L}\mathrm{tr}\left(P^{*}P+A_{i}^{*}A_{i}\right)=M$, and hence, ${\left||P|\right|}_{F}^{2}<M$. For any A∈A and $X=B_{1}P+B_{2}A$, the received pilot and data signals can be obtained from Y as follows: (4)$$B_{1}^{*}Y=PH+\sqrt{\frac{M}{\rho T}}B_{1}^{*}Z,$$
(5)$$B_{2}^{*}Y=AH+\sqrt{\frac{M}{\rho T}}B_{2}^{*}Z.$$

The matrix $B_{1}^{*}Y$ can then be used for channel estimation, and the matrix $B_{2}^{*}Y$ for data reception.

If $\left[B_{1}\;B_{2}\right]=I_{T}$, the PSAM is constructed by concatenating P and the matrices in A. Although these are the most extensively used values for $B_{1}$ and $B_{2}$, the (more general) definition of PSAM matrices in (Equation 3) provides a new interpretation of a PSAM, which will be used in Section 5 to design new constellations that combine coherent codes and USTMs. In addition to this, the orthogonal matrices $B_{1}$ and $B_{2}$ enable a two-dimensional graphical representation of the different constellations, as shown in Figure 1. In the figure, the (T−M)-dimensional subspace spanned by $B_{2}$, $S(B_{2})$, is represented in the horizontal axis, and the *M*-dimensional subspace spanned by $B_{1}$, $S(B_{1})$, in the vertical axis. The PSAM matrices can be viewed as shifts of the coherent code signals through the subspace spanned by $B_{1}$. As a result, these matrices have, in general, different energy, although this will ultimately depend on the coherent code. However, the USTM matrices have always the same energy, independently of how the USTM was constructed. In Figure 1, we highlight an interesting case in which the elements of a PSAM and a USTM span the same subspaces. In the following section, we will show that this case is not exceptional, i.e., it is always possible to find this type of PSAM-USTM pairs.

## 3. PSAM-USTM Equivalence

In this section, we will show that there is certain equivalence between PSAM and USTM constellations in terms of the spanned subspaces. We begin with the following result, which shows that, under very general conditions, the PSAM matrices span different subspaces.

**Theorem 1.** 
*Let P be invertible, and let the constellation $C^{P}={\left\{X_{i}^{P}\right\}}_{i=1}^{L}$ be constructed from A as in (Equation 3). Then, $A_{i}=A_{j}$ if and only if $S\left(X_{i}^{P}\right)=S\left(X_{j}^{P}\right)$.*


**Proof.** Assume that $A_{i}=A_{j}$. From (Equation 3), we have that $X_{i}^{P}=X_{j}^{P}$, and hence, $S\left(X_{i}^{P}\right)=S\left(X_{j}^{P}\right)$. Assume now that $S\left(X_{i}^{P}\right)=S\left(X_{j}^{P}\right)$. In this case, the columns of $X_{i}^{P}$ can be expressed as linear combinations of the columns of $X_{j}^{P}$, and vice versa. In particular, there is an M×M matrix R such that $X_{i}^{P}R=X_{j}^{P}$, which, pre-multiplying by $B_{1}^{*}$ and using (Equation 3), yields PR=P, and, since P is invertible, we have that $R=I_{M}$. Now, $B_{2}^{*}X_{i}^{P}RB_{2}^{*}X_{j}^{P}$ yields $A_{i}=A_{j}$. □

Theorem 1 shows that the subspaces spanned by PSAM matrices constructed from different coherent signals, $A_{i}$ and $A_{j}$, are different, and hence, that a PSAM could be decoded detecting the transmitted subspace, as a USTM receiver. The following corollary follows from Theorem 1.

**Corollary 1.** 
*Let $C^{P}$ be a PSAM constructed from A as in (Equation 3), where P is invertible. Then, there is a USTM, $C^{U}$, which elements span the same subspaces than the elements of $C^{P}$.*


In other words, any PSAM has an equivalent USTM in terms of the spanned subspaces. The elements of the USTM can be obtained from the elements of the PSAM by using any kind of orthonormalization. In particular, it is possible to find matrices ${\left\{R_{i}\right\}}_{i=1}^{L}$ whose inverses orthonormalize the elements of the PSAM. The equivalent USTM is, in this case, composed of the matrices
(6)$$X_{i}^{U}=X_{i}^{P}R_{i}^{-1},\;i=1,\ldots ,L.$$

The orthonormalization is, hence, a linear map. As an example, the matrices $X_{i}^{U}$ and $R_{i}$ can be obtained by means of a QR decomposition of $X_{i}^{P}$. It is important to note that the performance of the equivalent USTM does not depend on the particular orthonormalization. This fact is due to the following result.

**Proposition 1.** 
*Let the inverse of R orthonormalize $X\in C^{T\times M}$, T≥M. Then, the inverse of $\bar{R}$ orthonormalizes X if and only if $\bar{R}=UR$, for some squared unitary matrix U.*


**Proof.** Since R is invertible, $U=\bar{R}R^{-1}$ is well defined. The inverse of $\bar{R}$ orthonormalizes X if and only if
(7)$$\begin{array}{cc} I_{M}= & {\bar{R}}^{-1*}X^{*}X{\bar{R}}^{-1}=U^{-1*}R^{-1*}X^{*}XR^{-1}U^{-1}\\ = & U^{-1*}U^{-1}. \end{array}$$Therefore, U is unitary. □

Proposition 1 implies that any orthonormalization of X can be expressed using a given matrix R which inverse orthonormalizes X and a unitary matrix U. In this case and using (Equation 2), the received signal, if X is orthonormalized, is ${\mathrm{XR}}^{-1}U^{*}H+\sqrt{M/(\rho T)}Z$. Since the entries of H are i.i.d. drawn from CN(0,1), the entries of $U^{*}H$ follow exactly the same distribution. Using this fact, it can be shown that the knowledge of the particular orthonormalization, i.e., U, does not provide additional information on X. Therefore, the achievable rate is independent of U.

The equivalence pointed out in Corollary 1 also exists in the other direction, as stated in the following result.

**Theorem 2.** 
*Let $C^{U}={\left\{X_{i}^{U}\right\}}_{i=1}^{L}$ be a USTM and P an M×M invertible matrix. Then, there is a T×M unitary matrix $B_{1}$, with orthogonal complement $B_{2}$, such that $B_{1}^{*}X_{i}^{U}$, i=1,…,L, are invertible and ${\left\{\alpha X_{i}^{U}{\bar{R}}_{i}\right\}}_{i=1}^{L}$ is a PSAM which elements span the same subspaces than the elements of $C^{U}$, where*

(8)
$${\bar{R}}_{i}={\left(B_{1}^{*}X_{i}^{U}\right)}^{-1}P,$$


(9)
$$\alpha =\sqrt{\frac{LM}{\sum_{i=1}^{L}\left|\right|{\bar{R}}_{i}{\left|\right|}_{F}^{2}}}.$$


*In addition, if all the matrix signals in $C^{U}$ are equiprobable, then the PSAM satisfies the power constraint in (Equation 1).*


**Proof.** This proof is divided into four steps, in which we will prove that (i) matrix $B_{1}$, such that $B_{1}^{*}X_{i}^{U}$, i=1,…,L, are invertible, exists; (ii) ${\left\{\alpha X_{i}^{U}{\bar{R}}_{i}\right\}}_{i=1}^{L}$ is a PSAM; (iii) the elements of $C^{U}$ and ${\left\{\alpha X_{i}^{U}{\bar{R}}_{i}\right\}}_{i=1}^{L}$ span the same subspaces; and (iv) the constellation ${\left\{\alpha X_{i}^{U}{\bar{R}}_{i}\right\}}_{i=1}^{L}$ satisfies the power constraint in (Equation 1). The proofs of these steps will complete the proof of the theorem.(i) We start showing that $B_{1}$ exists such that $B_{1}^{*}X_{i}^{U}$, i=1,…,L, are invertible. To do this, we will use the following lemma.**Lemma 1.** ([22] (Exercise 14, p. 57))
*If ${\left\{X_{i}\right\}}_{i=1}^{L}$ are subspaces of equal dimension of a finite-dimensional vector space V over an infinite field F, then there is a subspace W of V such that the direct sum of W and each individual subspace in ${\left\{X_{i}\right\}}_{i=1}^{L}$ is V, that is, $V=W⊕X_{i}$, i=1,…,L. In other words, W is a common complement of the subspaces ${\left\{X_{i}\right\}}_{i=1}^{L}$.*We will use this lemma with the subspaces $W=S(B_{2})$ and $X_{i}=S\left(X_{i}^{U}\right)$, i=1,…,L. In particular, Lemma 1 implies that we can find $B_{2}$ such that $C^{T}=S\left(\left[X_{i}^{U}\;B_{2}\right]\right)$, i=1,…,L. We will use this result with a contradiction argument. Let us assume that we can find *i* such that $B_{2}^{*}X_{i}^{U}$ is not invertible. This implies that $S(X_{i}^{U})$ has, at least, one orthogonal direction to $S(B_{1})$, or, in other words, that there is a vector $X\in C^{M}$ such that $b^{*}B_{1}^{*}X_{i}^{U}X=0$ for all $b\in C^{M}$. Since $B_{2}$ is the orthogonal complement of $B_{1}$, then the direction orthogonal to $S(B_{1})$ is in the subspace spanned by this basis, i.e., $X_{i}^{U}X\in S\left(B_{2}\right)$. In this case, $X_{i}^{U}$ and $B_{2}$ cannot span $C^{T}$, which contradicts Lemma 1.(ii) In the second step, we will show that ${\left\{\alpha X_{i}^{U}{\bar{R}}_{i}\right\}}_{i=1}^{L}$ is a PSAM by showing that these matrices can be written as in (Equation 3). In particular, we will find the pilot matrix and the coherent code $A={\left\{A_{i}\right\}}_{i=1}^{L}$ to be used in (Equation 3). From (Equation 3), the pilot matrix used to construct the PSAM can be obtained by left-multiplying the PSAM matrices by $B_{1}^{*}$. In this case, this leads to
(10)$$\alpha B_{1}^{*}X_{i}^{U}{\bar{R}}_{i}=\alpha P,$$
for i=1,…,L. Now, using (Equation 3) again, the coherent code A can be obtained by left-multiplying the PSAM matrices by $B_{2}^{*}$, that is,
(11)$$A_{i}=\alpha B_{2}^{*}X_{i}^{U}{\bar{R}}_{i}=\alpha B_{2}^{*}X_{i}^{U}{\left(B_{1}^{*}X_{i}^{U}\right)}^{-1}P.$$The PSAM can then be constructed as in (Equation 3) using the pilot matrix αP and the coherent code A with elements in (Equation 11).(iii) Since the elements of ${\left\{\alpha X_{i}^{U}{\bar{R}}_{i}\right\}}_{i=1}^{L}$ are generated by scaling and right-multiplying by M×M invertible matrices the elements of the USTM, both span the same subspaces.(iv) If the transmitted signal, X, takes values from the constellation ${\left\{\alpha X_{i}^{U}{\bar{R}}_{i}\right\}}_{i=1}^{L}$ with equal probability, the average consumed energy in one channel block is
(12)$$E\left[{\left||X|\right|}_{F}^{2}\right]=\frac{\alpha^{2}}{L}\sum_{i=1}^{L}\left|\right|X_{I}^{U}{\bar{R}}_{I}{\left|\right|}_{F}^{2}=M,$$
where the second equality follows from (Equation 9) and the fact that $X_{i}^{U*}X_{i}^{U}=I_{M}$, i=1,…,L. Therefore, the power constraint in (Equation 1) is satisfied. □

Corollary 1 and Theorem 2 describe a PSAM-USTM equivalence in terms of the spanned subspaces. In particular, any PSAM has an equivalent USTM which elements span the same subspaces, and vice versa.

In the following section, we will use the results of this section to design a new USTM construction method. The resulting USTM constellations exhibit a certain structure inherited from a coherent code.

## 4. USTM Construction from PSAM Orthonormalization

In this section, we present a new USTM construction method based on the results in Section 3. In particular, a USTM, $C^{U}={\left\{X_{i}^{U}\right\}}_{i=1}^{L}$, can be constructed by othonormalizing the elements of a PSAM, $C^{P}={\left\{X_{i}^{P}\right\}}_{i=1}^{L}$, as in (Equation 6). In this case, the received signal when the *i*-th matrix of the constellation is transmitted results in
(13)$$Y=X_{i}^{U}H+\sqrt{\frac{M}{\rho T}}Z=X_{i}^{P}R_{i}^{-1}H+\sqrt{\frac{M}{\rho T}}Z.$$

The constellation construction is computationally simple, since it only requires an orthonormalization (by means of, e.g., a QR decomposition) in addition to the generation of the PSAM matrix. However, in general, the orthonormalization does not leave an apparent structure in $C^{U}$ that could be used to efficiently decode these signals. This fact implies that the optimum receiver should test all signals. In order to circumvent this issue, we note that the inverse of the $R_{i}$ matrix in (Equation 13) left-multiplies the channel matrix, but does not affect the noise. Therefore, it would be possible to interpret the $R_{i}^{-1}H$ term as an effective channel, and use a PSAM reception technique to detect $X_{i}^{P}$. It is important to note that this effective channel is not independent of the transmitted signal, which implies that using a PSAM reception technique is suboptimal. Despite this, we will show a case in which the combination of these transmission and reception techniques is better than directly transmitting the PSAM matrices. In particular, we tested this constellation design method with a PSAM constructed from a scaled identity pilot matrix and a coherent code whose matrix entries are independent and selected from a rectangular quadrature amplitude modulation (QAM), i.e., spatial multiplexing (SM) of QAM symbols. The available power was split between the pilot matrix and the coherent code following the indications in [1].

In Figure 2, we show the block error rate (BLER) for T=4, M=N=2, modulations of 4-QAM and 16-QAM, and three different transmitter-receiver configurations. The modulations of 4-QAM and 16-QAM lead to transmission rates of R=2 and R=4 bits per channel use (bpcu), respectively, whereas the transmitter-receiver configurations are detailed next:SM-SESD: In this case, the transmitter sent PSAM matrices constructed from a scaled identity pilot matrix and SM of QAM symbols. The receiver used a least squares (LS) channel estimator [23] and a Schnorr-Euchner sphere decoder (SESD) [24].SMO-SESD: This configuration is similar to the previous one. The only difference is that the signal matrices were orthonormalized before transmission. The receiver had no knowledge of this orthonormalization and assumed that the transmitted matrices were from the original PSAM constellation.SMO-ML: In this configuration, the transmitter orthonormalized the PSAM matrices too. The receiver tested all constellation matrices and performed a maximum likelihood (ML) detection [7].

The results in Figure 2 show that the SMO-ML configuration exhibits around 1 dB gain with respect to the other two configurations, albeit by means of using a computationally expensive receiver. In particular, since the SMO-ML receiver has to test all constellation matrices, the computational cost is proportional to the number of matrices in the constellation, i.e., $L=2^{RT}$, which follows an exponential law with respect to *R* and *T*. This fact makes the SMO-ML receiver extremely costly for relatively small values of *R* and/or *T*. As an illustrative example, for the cases drawn in Figure 2, the number of matrices are 256 and 65,536 for the R=2 and R=4 cases, respectively, and this value increases to 1,048,576 for R=5. The gain of the SMO-SESD configuration with respect to the SM-SESD is around 0.25 dB. In this case, the receivers are exactly the same, and the computational cost of the orthonormalization performed by the transmitter in the SMO-SESD configuration is polynomial with respect to *T* and *M*. Therefore, this gain is completely free for the receiver and relatively inexpensive for the transmitter.

The previous result shows that the orthonormalization of PSAM matrices is a simple idea that can provide some performance gain. In the rest of this section, we will examine how the orthonormalization modifies the structure of the PSAM and how this modification affects the performance. To do this, we graphically represented in Figure 3 the structures of two PSAMs and the equivalent USTMs using the $S(B_{1})$ and $S(B_{2})$ subspaces.

It can be easily inferred from the figure that the PSAM elements whose energy is lower than the constellation average, i.e., those inside the semicircle in Figure 3, are separated after the orthonormalization, and that the rest are bunched together. The separation between constellation elements is related to the protection against noise. In other words, Figure 3 suggests that the orthonormalization is beneficial for the PSAM elements with less energy than the average, but disadvantageous for the rest.

We verified this effect in the constellations of Figure 2. In particular, we studied how the PSAM matrix energy affects the BLER of the SM-SESD and SMO-SESD transmitter-receiver configurations with R=4. This result is depicted in Figure 4, where the energy of the *i*-th PSAM matrix was computed as $\left|\right|X_{i}^{P}{\left|\right|}_{F}^{2}$. The figure confirms that the orthonormalization is beneficial below certain energy of the PSAM matrix.

Looking at Figure 3, it is possible to identify a better USTM in which the bunched elements are stretched out towards the $S(B_{2})$ subspace. Unfortunately, this type of USTMs do not correspond to a simple orthonormalization of the elements of a PSAM. In the following section, we will propose another USTM constellation construction method to circumvent the limits of the PSAM orthonormalization.

## 5. USTM Construction Combining a Coherent Code and a Small USTM

### 5.1. General Description

In this section, we present another USTM construction method that is inspired by the definition of PSAM matrices in (Equation 3), and based on the combination and orthonormalization of a USTM with $L^{U}$ elements and a coherent code of $L^{P}$ elements to generate a constellation of $L=L^{U}L^{P}$ elements. The main idea is to compose $L^{U}$ PSAMs using the USTM elements to carry the pilot matrix in place of the $B_{1}$ matrix in (Equation 3). In particular, letting $C^{U}={\left\{X_{i}^{U}\right\}}_{i=1}^{L^{U}}$, $C^{U⊥}={\left\{X_{i}^{U⊥}\right\}}_{i=1}^{L^{U}}$, P, and $A={\left\{A_{i}\right\}}_{i=1}^{L^{P}}$ be the USTM, a set of orthogonal complements of the USTM elements, a pilot matrix, and the coherent code, respectively, we propose to compose the PSAMs $C_{i}^{P}={\left\{X_{i,j}^{P}\right\}}_{j=1}^{L^{P}}$, $i=1,\ldots ,L^{U}$, where
(14)$$X_{i,j}^{P}=X_{i}^{U}P+X_{i}^{U⊥}A_{j}.$$

Subsequently, we use these PSAMs to construct the constellation $C^{O}={\left\{{\left\{X_{i,j}^{O}\right\}}_{i=1}^{L^{U}}\right\}}_{j=1}^{L^{P}}$, where $X_{i,j}^{O}$ is the orthonormalization of $X_{i,j}^{P}$, i.e., we can find a matrix $R_{i,j}$ such that $X_{i,j}^{O}=X_{i,j}^{P}R_{i,j}^{-1}$ is unitary. Following the general PSAM description of Section 2, we assume that the PSAMs ${\left\{C_{i}^{P}\right\}}_{i=1}^{L^{U}}$ satisfy the power constraint in (Equation 1), i.e., the pilot matrix power $P_{P}={\left||P|\right|}_{F}^{2}$ and the coherent code power $P_{A}=\frac{1}{L^{P}}\sum_{j=1}^{L^{P}}\left|\right|A_{j}{\left|\right|}_{F}^{2}$ satisfy $P_{P}+P_{A}=M$.

A two-dimensional graphical representation of this constellation construction method is depicted in Figure 5. The figure illustrates one of the benefits of this method: the constellation elements are more homogeneously distributed in the semicircle than in Figure 3. In addition to a composition of $L^{U}$ orthonormalized PSAMs, the constellation $C^{O}$ admits another interpretation. In particular, it can be seen as a generalization of a PSAM orthonormalization in which data are encoded also in the pilots, or, more specifically, in the location of the pilot matrix within a channel block. This location is given by the USTM matrices in $C^{U}$. It is important to note that, assuming $L^{U}$ is small enough, both constructing $C^{U}$ using numerical optimization tools and storing $C^{U}$ and $C^{U⊥}$ is feasible.

The receiver can estimate the transmitted signal using a PSAM and a USTM receiver in two phases. In the first phase, we propose to select one candidate from each PSAM $C_{i}^{P}$, $i=1,\ldots ,L^{U}$, using the PSAM receiver. More specifically, for the *i*-th PSAM, $C_{i}^{P}$, the PSAM receiver should assume that the *i*-th matrix from $C^{U}$ was used to carry the pilot matrix and estimate the transmitted coherent matrix, say, $A_{{\hat{j}}_{i}}$. In the second phase, the USTM receiver assumes that the constellation used by the transmitter was the set of candidates, i.e., ${\left\{X_{i,{\hat{j}}_{i}}^{O}\right\}}_{i=1}^{L^{U}}$, and selects the received matrix from this set. The computational cost of this reception technique is equivalent to the cost of $L^{U}$ PSAM receivers for constellations of $L^{P}$ elements, plus the cost of a USTM receiver for constellations of $L^{U}$ elements.

Apart from the power constraint, this construction method does not impose any other limitation on how the available power is distributed between P and A. However, this distribution affects the performance of $C^{O}$, and, hence, it should be carefully designed. In the following two sections, we will propose two methods to select $P_{P}$ and $P_{A}$. The first method, in Section 5.2, is a general approach that can be used for any system, although it is very conservative and has a limited performance, especially for certain coherent codes. The second method, in Section 5.3, is valid for certain *T*, *M*, and $L^{U}$ values, and the generated constellations present some performance gains as compared with other constellation constructions.

### 5.2. General Method to Select $P_{P}$ and $P_{A}$

Although the authors of [1] obtained the optimal power distribution between the pilot matrix and the coherent code for one PSAM, the combination of several PSAMs to compose a unique constellation limits the performance of this power distribution. In particular, the elements of the resulting constellation $C^{O}$ are intermingled, and even some of them could span the same subspace, as depicted in Figure 6. In order to avoid this, the distance between the subspaces spanned by the constellation elements has to be controlled. For instance, let $d_{S}\left(X_{1},X_{2}\right)$ be the distance between the subspaces spanned by the matrices $X_{1}$ and $X_{2}$, if $P_{P}$ and $P_{A}$ are selected in such a way that
(15)$$\max_{i,j}d_{S}\left(X_{i}^{U},X_{i,j}^{O}\right)<\frac{1}{2}\min_{k,\ell \neq k}d_{S}\left(X_{k}^{U},X_{\ell }^{U}\right),$$
then it can be shown that the constellation elements span different subspaces. In particular, using (Equation 15) and the triangle inequality, i.e., $d_{S}\left(X_{1},X_{3}\right)\leq d_{S}\left(X_{1},X_{2}\right)+d_{S}\left(X_{2},X_{3}\right)$, we have that
(16)$$\begin{array}{cc} d_{S}\left(X_{i}^{U},X_{k}^{U}\right)\;\leq & d_{S}\left(X_{i}^{U},X_{i,j}^{O}\right)\;+\;d_{S}\left(X_{i,j}^{O},X_{k,\ell }^{O}\right)\;+\;d_{S}\left(X_{k}^{U},X_{k,\ell }^{O}\right)\\ < & d_{S}\left(X_{i}^{U},X_{k}^{U}\right)\;+\;d_{S}\left(X_{i,j}^{O},X_{k,\ell }^{O}\right), \end{array}$$
for all $i=1,\ldots ,L^{U}$, $k=1,\ldots ,i-1,i+1,\ldots ,L^{U}$, $j=1,\ldots ,L^{P}$, and $\ell =1,\ldots ,L^{P}$. The inequality in (Equation 16) implies that $d_{S}\left(X_{i,j}^{O},X_{k,\ell }^{O}\right)>0$ and, hence, that $X_{i,j}^{O}$ and $X_{k,\ell }^{O}$ span different subspaces. It is important to note that the previous result is independent of the particular metric used to measure the subspace distance.

The main drawback of this method is that it does not take into account the shape of the coherent code, A, or the orientation of the orthogonal complements in $C^{U⊥}$. In fact, this method ensures that the constellation elements span different subspaces even in cases in which a bad combination of coherent code and orthogonal complements is used. This implies that this method is, in general, conservative and not very efficient. Figure 7 illustrates this drawback with two examples. In particular, the example in Figure 7a shows three PSAMs constructed from a square-shaped coherent code, which power has to be reduced to avoid that the elements near the square corners intersect with those of the other PSAMs. This power level generates a lot of empty space, although, in this case, the orthonormalization of the elements in Figure 7a satisfy (Equation 15). In Figure 7b, the orthogonal complements are rotated $45^{\circ }$, which allows more power to be allocated to the coherent code. In this case, the large empty spaces are eliminated and the constellation elements are more homogeneously distributed, although their orthonormalization does not necessarily satisfy (Equation 15).

In the following section, we will describe a method to produce constellations similar to that depicted in Figure 7b. In particular, prior to the orthonormalization, the constellation elements are located in the surface of a hypercube defined in a special vector set: a module over a ring.

### 5.3. USTM Construction from a Hypercube

In this section, we present a USTM construction method in which the constellation elements are first placed in the surface of a hypercube, and then they are orthonormalized. Hypercubes are generally defined in vector spaces over the field of the real numbers R, i.e., $R^{n}$. However, for this construction method, we need to define the hypercube in a different vector set in which the field R is substituted by a ring. This type of vector sets are known as modules over a ring. In this section, (i) we present the module in which the hypercube is defined, (ii) we introduce a hypercube definition that can be used in this module, and (iii) we describe the method itself.

Assuming that *T* is a multiple of *M*, i.e., $\frac{T}{M}$ is an integer, we can interpret $C^{T\times M}$ as the Cartesian product of $\frac{T}{M}$ sets $R=C^{M\times M}$, i.e., $M=C^{T\times M}=R^{\frac{T}{M}}$. It is important to note that R is not commutative and, hence, it is not a field, which implies that M is not a vector space over R. However, it can be easily verified that R is a ring equipped with the operations of matrix sum and product. Consequently, M is a module over R. More specifically, M is a right R-module equipped with the matrix product between elements of the two sets. Note that, using this interpretation, X∈M is a vector with $\frac{T}{M}$ components, and that each component is a matrix in R. In addition to this, this module has a basis, which can be built from any T×T unitary matrix. In particular, let $B={\left\{B_{i}\right\}}_{i=1}^{\frac{T}{M}}$ be a collection of T×M matrices such that $\left[B_{1}\;\cdots \;B_{\frac{T}{M}}\right]$ is unitary, i.e.,
(17)$$\left[B_{1}\;\cdots \;B_{\frac{T}{M}}\right]{\left[B_{1}\;\cdots \;B_{\frac{T}{M}}\right]}^{*}=\sum_{i=1}^{\frac{T}{M}}B_{i}B_{i}^{*}=I_{T},$$
then B is a basis of M. In other words, for any X∈M, we can find coordinates $C_{i}\in R$, $i=1,\ldots ,\frac{T}{M}$, such that $X=\sum_{i=1}^{\frac{T}{M}}B_{i}C_{i}$. In particular, it can be shown from (Equation 17) that the matrices $C_{i}=B_{i}^{*}X$ satisfy the previous equality. Since the elements of B also satisfy $B_{i}^{*}B_{j}=0_{M}$, for all i≠j, and $B_{i}^{*}B_{i}=I_{M}$, for all *i*, where $0_{M}$ is the M×M null matrix, we say that B is an orthonormal basis.

We define the hypercube in M defined by the orthonormal basis B as the set
(18)$$H_{B}\left(\xi \right)=\left\{X=\sum_{i=1}^{\frac{T}{M}}B_{i}C_{i}\left||\right|C_{i}{\left|\right|}_{F}\leq \xi ,\forall i\right\},$$
where ξ>0 is a constant that defines the size of the hypercube. It can be readily verified that, in the case R=R, $H_{B}\left(\xi \right)$ is an hypercube centered at the origin, oriented following the elements in the basis B, and whose sides expand from −ξ to ξ in the coordinate system defined by B. The surface of the hypercube is the set of points in which the constraint in (Equation 18) is satisfied as an equality for at least one matrix $C_{i}$, i.e., the surface of $H_{B}\left(\xi \right)$ is
(19)$$H_{B}^{S}\left(\xi \right)=\left\{X\in H_{B}\left(\xi \right)\left|\exists i,|\right|B_{i}^{*}X{\left|\right|}_{F}=\xi \right\}.$$

We say that all the matrices X that satisfy $||B_{i}^{*}X{\left|\right|}_{F}=\xi$ for the same coordinate *i* are in the same face of the hypercube.

After the introduction of the hypercube definition, we are able to describe the construction of the USTM. To do that, we start composing $\frac{T}{M}$ PSAMs with elements in different faces of a hypercube. These PSAMs are composed using an orthonormal basis B of the module M, a coherent code $A^{H}={\left\{A_{i}^{H}\right\}}_{i=1}^{L^{P}}$, where $A_{i}^{H}\in C^{M\times M}$, and a pilot matrix P. Note that, in this case, the coherent code elements are drawn from $C^{M\times M}$ and not from $C^{T-M\times M}$ as before. The reason for this is that the coherent code elements are going to be used as coordinates of matrices in M. Using the previous pieces, we can compose the PSAMs $C_{i}^{\mathrm{HP}}={\left\{X_{i,\iota }^{\mathrm{HP}}\right\}}_{\iota \in I}$, $i=1,\ldots ,\frac{T}{M}$, where $\iota ={\left[\iota_{1}\cdots \iota_{\frac{T}{M}-1}\right]}^{*}$ is a vector of indices, $I={\left\{1,\ldots ,L^{P}\right\}}^{\frac{T}{M}-1}$,
(20)$$X_{i,\iota }^{\mathrm{HP}}=\mu \sum_{j=1}^{i-1}B_{j}A_{\iota_{j}}^{H}+B_{i}P+\mu \sum_{j=i+1}^{\frac{T}{M}}B_{j}A_{\iota_{j-1}}^{H},$$
and μ>0 is a factor that scales the coherent code. As mentioned before, it is clear from (Equation 20) that $X_{i,\iota }^{\mathrm{HP}}$ has coordinates in the module M that are scaled versions of the elements of the coherent code $A^{H}$, except for the *i*-th coordinate, which is the pilot matrix. It can be shown that, with a proper μ value, the PSAM elements are in different faces of the hypercube $H_{B}\left({\left||P|\right|}_{F}\right)$. In particular, μ has to satisfy
(21)$$\mu ||A_{i}^{H}{\left|\right|}_{F}{<||P||}_{F},$$
for all $i=1,\ldots ,L^{P}$. Note that the strict inequality is required to ensure that the PSAM elements are not placed in two hypercube faces at the same time. We now use the PSAMs $C_{i}^{\mathrm{HP}}$, $i=1,\ldots ,\frac{T}{M}$, to construct the constellation $C^{\mathrm{HO}}={\left\{{\left\{X_{i,\iota }^{\mathrm{HO}}\right\}}_{\iota \in I}\right\}}_{i=1}^{\frac{T}{M}}$, where $X_{i,\iota }^{\mathrm{HO}}$ is the orthonormalization of $X_{i,\iota }^{\mathrm{HP}}$. The number of elements in this constellation is $\frac{T}{M}{\left(L^{P}\right)}^{\frac{T}{M}-1}$.

The following result ensures that the elements in $C^{\mathrm{HO}}$ span different subspaces.

**Theorem 3.** 
*Let the pilot matrix used to compose $C^{\mathrm{HO}}$ be a scaled unitary matrix, i.e., P=ηU, where η>0 and U is unitary; and let $X_{i,\iota }^{\mathrm{HO}}\in C^{\mathrm{HO}}$ and $X_{j,\kappa }^{\mathrm{HO}}\in C^{\mathrm{HO}}$. Then, $S\left(X_{i,\iota }^{\mathrm{HO}}\right)=S\left(X_{j,\kappa }^{\mathrm{HO}}\right)$ if and only if i=j and ι=κ.*


**Proof.** We start by noting that $X_{i,\iota }^{\mathrm{HO}}$ and $X_{i,\iota }^{\mathrm{HP}}$ span the same subspaces. Therefore, it is equivalent to show that $S\left(X_{i,\iota }^{\mathrm{HP}}\right)=S\left(X_{j,\kappa }^{\mathrm{HP}}\right)$ if and only if i=j and ι=κ. Using Theorem 1, it is straightforward to prove this theorem for the case i=j. Therefore, we continue assuming i≠j. We will proceed by assuming that $S\left(X_{i,\iota }^{\mathrm{HP}}\right)=S\left(X_{j,\kappa }^{\mathrm{HP}}\right)$, and conclude that, in this case, the inequality in (Equation 21) cannot be satisfied, which concludes the proof. In particular, if $S\left(X_{i,\iota }^{\mathrm{HP}}\right)=S\left(X_{j,\kappa }^{\mathrm{HP}}\right)$, there is some $R\in C^{M\times M}$ such that $X_{i,\iota }^{\mathrm{HP}}=X_{j,\kappa }^{\mathrm{HP}}R$. Pre-multiplying both sides of this equality by $B_{i}^{*}$ and $B_{j}^{*}$, we obtain P=μDR and μE=PR, respectively, for some $D,E\in A^{H}$. Therefore, since P is invertible, then all D, E and R are invertible. Moreover, we can combine both equalities to obtain $\mu E=\frac{1}{\mu }PD^{-1}P=\frac{\eta^{2}}{\mu }UD^{-1}U$. Both D and E must satisfy the inequality in (Equation 21), and thus, we should have that
(22)$$\mu^{2}\mathrm{tr}\left(D^{*}D\right)<\mathrm{tr}\left(P^{*}P\right)=\eta^{2}M,$$
(23)$$\begin{array}{cc} \mu^{2}\mathrm{tr}\left(E^{*}E\right)= & \frac{1}{\mu^{2}}\mathrm{tr}\left(P^{*}{D^{*}}^{-1}P^{*}PD^{-1}P\right)\\ = & \frac{\eta^{4}}{\mu^{2}}\mathrm{tr}\left({D^{*}}^{-1}D^{-1}\right)<\eta^{2}M. \end{array}$$Let $\lambda_{1},\ldots ,\lambda_{M}$ be the eigenvalues of D. The two previous inequalities imply that
(24)$$\mu^{2}\sum_{m=1}^{M}\lambda_{m}<\eta^{2}M,$$
(25)$$\frac{\eta^{4}}{\mu^{2}}\sum_{m=1}^{M}\frac{1}{\lambda_{m}}<\eta^{2}M.$$For a given sum of the eigenvalues, the left-hand side of (Equation 25) is minimum if all the eigenvalues are equal. Therefore,
(26)$$\frac{\eta^{4}}{\mu^{2}}\sum_{m=1}^{M}\frac{1}{\lambda_{m}}>\frac{\eta^{4}}{\mu^{2}}\sum_{n=1}^{M}\frac{M}{\sum_{m=1}^{M}\lambda_{m}}=\frac{\eta^{4}}{\mu^{2}}\frac{M^{2}}{\sum_{m=1}^{M}\lambda_{m}}.$$Using (Equation 24) in (Equation 26) we obtain
(27)$$\frac{\eta^{4}}{\mu^{2}}\sum_{m=1}^{M}\frac{1}{\lambda_{m}}>\eta^{2}M,$$
which contradicts (Equation 25). □

As a consequence of Theorem 3, if P is a scaled unitary matrix and μ satisfies (Equation 21) for all the elements in the coherent code $A^{H}$, then the elements in $C^{\mathrm{HO}}$ span different subspaces. This fact highlights the importance of a good selection of the scaling factor μ. In addition to this, it is important to note that μ significantly impacts the performance of the constellation $C^{\mathrm{HO}}$. In order to visualize this impact, we refer to Figure 7b, which is a graphical representation of the PSAMs $C_{i}^{\mathrm{HP}}$, $i=1,\ldots ,\frac{T}{M}$, for the case $\frac{T}{M}=3$. Note that the ’×’ symbols in one of the faces of the cube in Figure 7b represent one constellation element, and that the coherent code $A^{H}$ used for this graphical representation is composed of four elements. The scaling factor μ controls how the elements of each PSAM are distributed throughout the cube faces. In particular, a low value causes the PSAM elements to be concentrated around the axes, whereas a large value makes the elements approach the face borders, and thus, the elements in a neighbouring cube face. In order to obtain a good performance, we have to set the value of μ taking into account the protection against noise and the reception technique.

Regarding the protection against noise, we should equalize the elements’ separation in each PSAM, and the separation between elements in different PSAMs. To this end, we propose to select μ in such a way that
(28)$$\min_{i\neq j}d_{F}\left(\mu A_{i}^{H},\mu A_{j}^{H}\right)=\sqrt{2}\min_{i,Q}d_{F}\left(\mu A_{i}^{H},Q\right),$$
where $d_{F}\left(X,Y\right)$ is the Euclidean distance between X and Y, i.e., $d_{F}\left(X,Y\right)=||X-{Y||}_{F}$, and Q is restricted to ${\left||Q|\right|}_{F}={\left||P|\right|}_{F}$. The left-hand side of (Equation 28) is the minimum distance between two coherent code elements, which is, at the same time, the minimum distance between two elements of the same PSAM. The minimization in the right hand side of (Equation 28) provides the minimum distance between the coherent code elements and a matrix with Frobenius norm equal to ${\left||P|\right|}_{F}$. Figure 8 provides a graphical representation of the previous distances for a $\frac{T}{M}=2$ case.

Regarding the reception technique, if a PSAM receiver is used to select candidates from the PSAMs, we should consider the power split between the pilot matrix and the coherent code obtained in [1]. In this case, we propose to set μ to the minimum value between the one that satisfies (Equation 28), and the one that satisfies the power split in [1].

In the following section, we will use the hypercube method described in this section to obtain USTM constellations from coherent codes composed of matrices with rectangular QAM symbols in their entries.

### 5.4. Application to Rectangular QAM-Based Coherent Codes

In this section, we focus on a special case of the hypercube method to construct USTM constellations. In particular, the coherent codes of this case are based on rectangular QAM, i.e., the real and imaginary parts of each matrix entry in the coherent codes are independently and equiprobably drawn from an equispaced set of values centered at the origin. More specifically, the set of values is
(29)$$Q={\left\{q\left(i-\frac{L^{Q}+1}{2}\right)\right\}}_{i=1}^{L^{Q}},$$
where $L^{Q}$ is the number of elements in Q, and q>0 is a factor that scales the QAM constellation. In the rest of this section, we assume that a PSAM receiver is used to select candidates from the PSAMs, and we obtain the value of μ following the method exposed in the previous section.

We start finding the value of μ that satisfies (Equation 28), which we express as $\mu_{1}$. To this aim, we note that, for coherent codes based on rectangular QAM, the left-hand side of (Equation 28) is $\mu_{1}q$. In order to find the right-hand side of (Equation 28), we first solve
(30)$$\begin{array}{cc} \min_{Q} & \;d_{F}^{2}\left(X,Q\right),\\ s.t., & {\;||Q||}_{F}^{2}=\xi^{2}. \end{array}$$

The Lagrangian of (Equation 30) is $L\left(Q\right)=\mathrm{tr}\left({\left(X-Q\right)}^{*}\left(X-Q\right)\right)+\sigma \left(\mathrm{tr}\left(Q^{*}Q\right)-\xi^{2}\right)$, where σ is a Lagrange multiplier. Using the differential of the Lagrangian, it can be shown that the matrix Q that solves (Equation 30) must satisfy −+σ=0, which yields $Q=\frac{1}{1+\sigma }X$. This implies that the solution of (Equation 30) is proportional to the given matrix X. Using these results, the equality in (Equation 28) can be expressed as
(31)$$\begin{array}{cc} \mu_{1}q= & \sqrt{2}\min_{i}d_{F}\left(\mu_{1}A_{i}^{H},\frac{{\left||P|\right|}_{F}}{\left|\right|A_{i}^{H}{\left|\right|}_{F}}A_{i}^{H}\right)\\ = & \sqrt{2}\min_{i}|\mu_{1}-\frac{{\left||P|\right|}_{F}}{\left|\right|A_{i}^{H}{\left|\right|}_{F}}\left||\right|A_{i}^{H}{\left|\right|}_{F}. \end{array}$$

Since $\mu_{1}$ must satisfy (Equation 21), we have that
(32)$$\mu_{1}<\frac{{\left||P|\right|}_{F}}{\left|\right|A_{i}^{H}{\left|\right|}_{F}},$$
and hence, the equality in (Equation 31) results in
(33)$$\begin{array}{cc} \mu_{1}q= & \sqrt{2}\min_{i}\left(\frac{{\left||P|\right|}_{F}}{\left|\right|A_{i}^{H}{\left|\right|}_{F}}-\mu_{1}\right)\left|\right|A_{i}^{H}{\left|\right|}_{F}\\ = & \sqrt{2}\min_{i}\left({\left||P|\right|}_{F}-\mu_{1}\left|\right|A_{i}^{H}{\left|\right|}_{F}\right). \end{array}$$

The coherent code matrix that minimizes the last term in (Equation 33) is the one with the maximum norm. For the case of rectangular QAM, we have that $\max_{i}\left|\right|A_{i}^{H}{\left|\right|}_{F}=\sqrt{2}M\frac{L^{Q}-1}{2}q$. Using this result in (Equation 33) yields
(34)$$\mu_{1}q=\sqrt{2}{\left||P|\right|}_{F}-M\left(L^{Q}-1\right)\mu_{1}q.$$

Consequently,
(35)$$\mu_{1}=\frac{{\left||P|\right|}_{F}}{q}\frac{\sqrt{2}}{1+M(L^{Q}-1)}.$$

We continue computing the value of μ that satisfies the power split in [1], which we express as $\mu_{2}$. This power split is given in [1] in terms of the portion of power dedicated to the coherent code,
(36)$$\alpha =\frac{P_{A}}{P_{A}+P_{P}}=\frac{P_{A}}{P_{A}+{\left||P|\right|}_{F}^{2}}.$$

Since all the QAM symbols are equiprobable, the average power used in the coherent code is
(37)$$\begin{array}{cc} P_{A}= & \frac{2(T-M)M}{L^{Q}}\sum_{i=1}^{L^{Q}}\mu_{2}^{2}q^{2}{\left(i-\frac{L^{Q}+1}{2}\right)}^{2}\\ = & 2\left(T-M\right)M\mu_{2}^{2}q^{2}\frac{{\left(L^{Q}\right)}^{2}-1}{12}. \end{array}$$

Using (Equation 36) and (Equation 37), we obtain
(38)$$\mu_{2}=\frac{{\left||P|\right|}_{F}}{q}\sqrt{\frac{\alpha }{1-\alpha }\frac{1}{(T-M)M}\frac{6}{{\left(L^{Q}\right)}^{2}-1}},$$
where, as indicated in [1], $\alpha =\frac{1}{2}$ if T=2M and $\alpha =\gamma -\sqrt{\gamma (\gamma -1)}$ if T>2M, where $\gamma =\frac{\left(T-M)(M+\rho T\right)}{\rho T(T-2M)}$.

Finally, as proposed in the previous section, we scale the coherent code by $\mu =\min(\mu_{1},\mu_{2})$. In the following section, we compare the performance of these constellations with the performance of a PSAM and its orthonormalization.

## 6. Constellation Performance Analysis

In this section, we analyze and compare the performance of three transmitter-receiver configurations. Two of those configurations were briefly analyzed in Section 4. In particular, the three configurations are as follows:SM-SESD: The transmitter sends PSAM matrices constructed from a scaled identity pilot matrix and SM of QAM symbols. The receiver uses an LS channel estimator and a SESD.SMO-SESD: The same configuration as the previous one, but the matrices are orthonormalized before transmission.H-SESD-ML: In this configuration, the transmitter uses a constellation generated using the hypercube method with an identity pilot matrix and rectangular QAM-based coherent codes, as described in Section 5.3 and Section 5.4. The receiver uses the reception technique described in Section 5.1, i.e., it estimates the transmitted signal using a PSAM and a USTM receiver in two phases. In the first phase, the receiver selects one candidate from each PSAM. To this end, it assumes that the transmitted signal belongs to a given PSAM and uses a LS channel estimator and a SESD to estimate the transmitted signal (one for each PSAM). In the second phase, the USTM receiver assumes that the constellation used by the transmitter was the set of candidates and selects the received matrix from this set using an ML detection.

We first compare the BLER of these configurations for M=N=2, *T* equal to 4, 8 and 16, and for three QAM sizes: 4-QAM, 16-QAM, and 64-QAM. The BLER is depicted in Figure 9, Figure 10 and Figure 11 for T=4, T=8, and T=16, respectively. The figures show that the performance of the three configurations is very similar. SMO-SESD is slightly better than SM-SESD and the performance of H-SESD-ML with respect to the other two configurations depends on the particular case. However, it is important to note that the constellation used in the H-SESD-ML configuration has more elements than those used in the other configurations. In particular, the constellations of the SM-SESD and SMO-SESD have ${\left(L^{Q}\right)}^{2M(T-M)}$ elements, and the constellation used in the H-SESD-ML configuration has $\frac{T}{M}{\left(L^{Q}\right)}^{2M(T-M)}$ elements, where $L^{Q}$ equals 2, 4 and 8 for 4-QAM, 16-QAM and 64-QAM, respectively. As a consequence, the bits transmitted per channel use are different for each configuration, as shown in Table 1. Therefore, the BLER obtained with the H-SESD-ML configuration cannot be directly compared.

In order to perform a fairer comparison, we computed the effective throughput of the three configurations as the nominal throughput in Table 1 times one minus the BLER in Figure 9, Figure 10 and Figure 11. The maximum of the effective throughput from modulations 4-QAM, 16-QAM and 64-QAM is shown in Figure 12. Mathematically, letting $R_{i}$ and ${\mathrm{BLER}}_{i}\left(\rho \right)$ be the nominal throughput and the BLER for an SNR ρ of the *i*-th modulation, respectively, the effective throughput $R_{\mathrm{eff}}=\max_{i}R_{i}\left(1-{\mathrm{BLER}}_{i}\left(\rho \right)\right)$ is shown in Figure 12. The figure shows that H-SESD-ML is the best configuration for T=8 and T=16, and it is also the best configuration for some SNR ranges for T=4. With respect to SM-SESD, SMO-SESD increases the effective throughput around 0.04 bpcu on average for the three values of the channel block length *T*, whereas the increment provided by H-SESD-ML is around 0.06 bpcu for T=4, 0.2 bpcu for T=8 and 0.17 bpcu for T=16.

## 7. Conclusions

We studied the relationship between PSAMs and USTMs. To do this, we introduced a new definition of PSAM space-time matrices, in which the pilot matrix can be spread out across the coherence interval using a tall unitary matrix. This unitary matrix and its orthogonal complement enables a graphical representation of these constellations. Using this definition, we found a map that transforms any PSAM into a USTM and vice versa. This map indicates a PSAM-USTM equivalence in terms of the spanned subspaces.

We used this equivalence and the graphical representation to define new USTM construction methods based on different PSAMs and their equivalent USTMs. In particular, firstly, we proposed the use of the equivalent USTM of a PSAM. Secondly, inspired by the graphical representation, we proposed to combine the equivalent USTMs of several PSAMs of which the pilots are carried by different tall unitary matrices. The resulting constellations can be interpreted as a PSAM where information is also encoded in the pilot position inside the channel coherence interval. Using the correct power distribution between pilots and signals, our results show that the obtained USTMs have higher effective throughput than the PSAMs used to construct them.

## Figures and Tables

**Figure 1 sensors-22-09049-f001:**
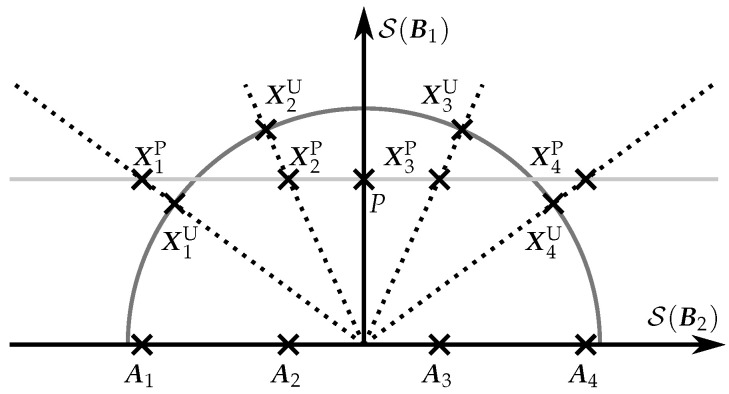
Graphical representation of a coherent code $A={\left\{A_{i}\right\}}_{i=1}^{4}$, a PSAM $C^{P}={\left\{X_{i}^{P}\right\}}_{i=1}^{4}$, and a USTM $C^{U}={\left\{X_{i}^{U}\right\}}_{i=1}^{4}$. The signals of both the PSAM and the USTM have the same mean power and span the same subspaces, which are drawn with dotted lines.

**Figure 2 sensors-22-09049-f002:**
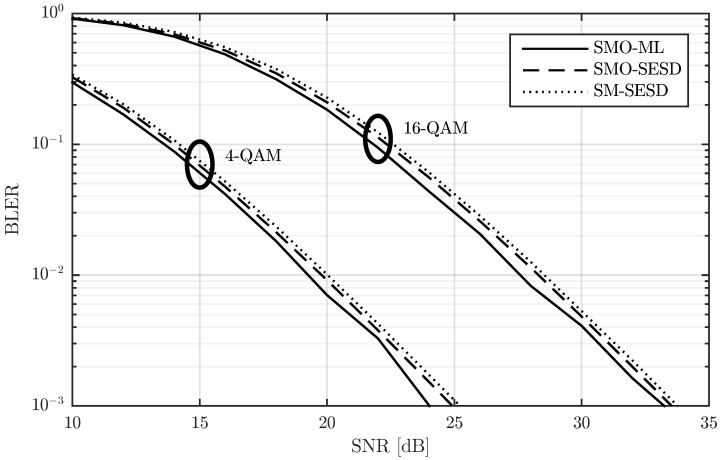
Performance comparison of a PSAM and its element orthonormalization with two different receivers for T=4, M=N=2, and two transmission rates.

**Figure 3 sensors-22-09049-f003:**
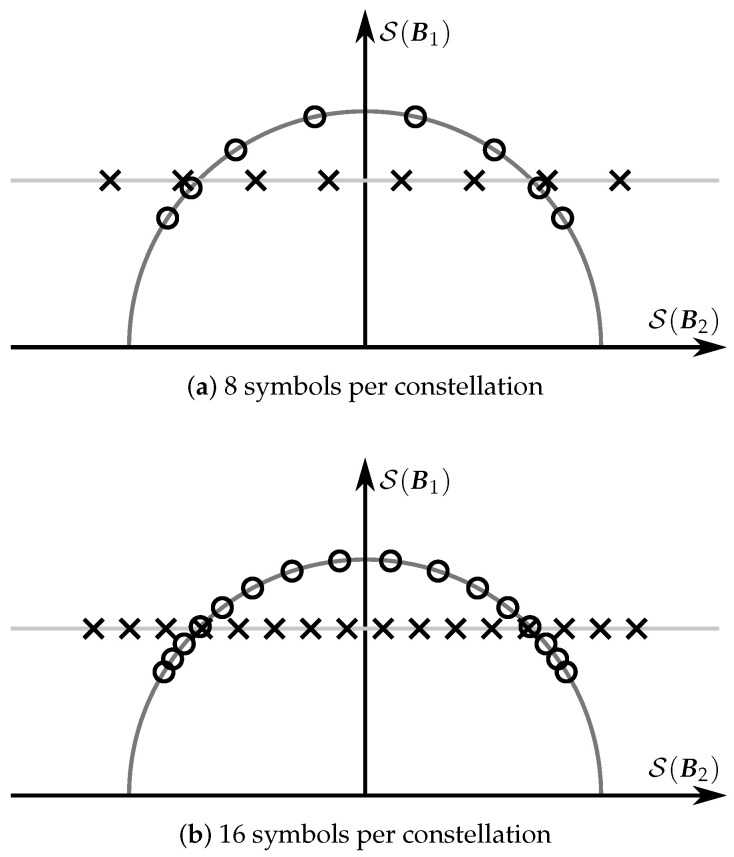
Two-dimensional graphical representation of the elements of a PSAM (with ‘×’ symbols) and the equivalent USTM (with ‘∘’ symbols).

**Figure 4 sensors-22-09049-f004:**
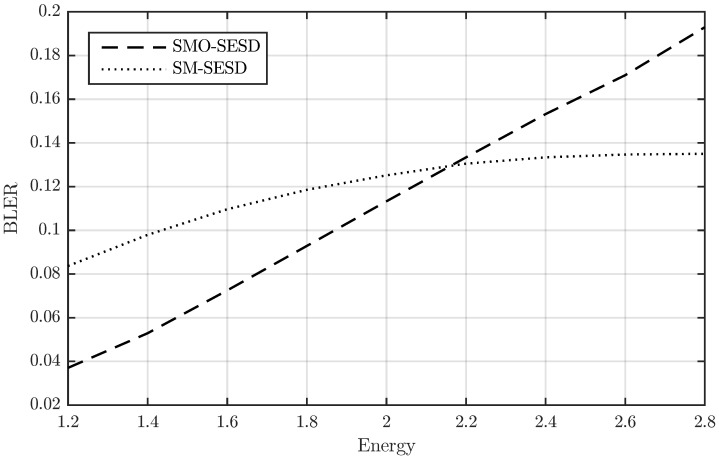
BLER vs. PSAM matrix energy of a PSAM and its element orthonormalization for T=4, M=N=2, R=4, and an SNR of 22 dB.

**Figure 5 sensors-22-09049-f005:**
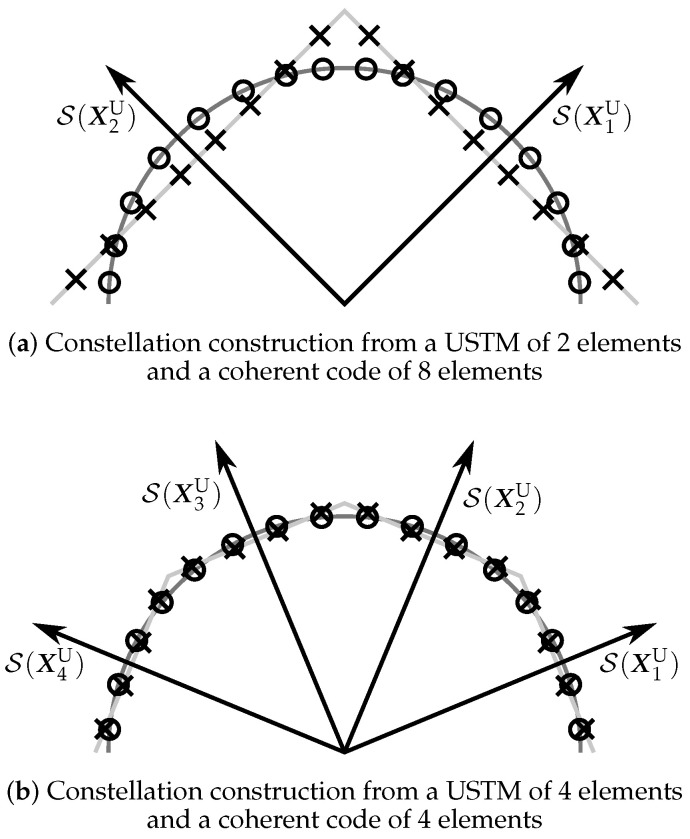
Two-dimensional graphical representation of the elements of the PSAMs constructed combining a USTM and a coherent code (with ‘×’ symbols), and the orthonormalization of these elements (with ‘∘’ symbols).

**Figure 6 sensors-22-09049-f006:**
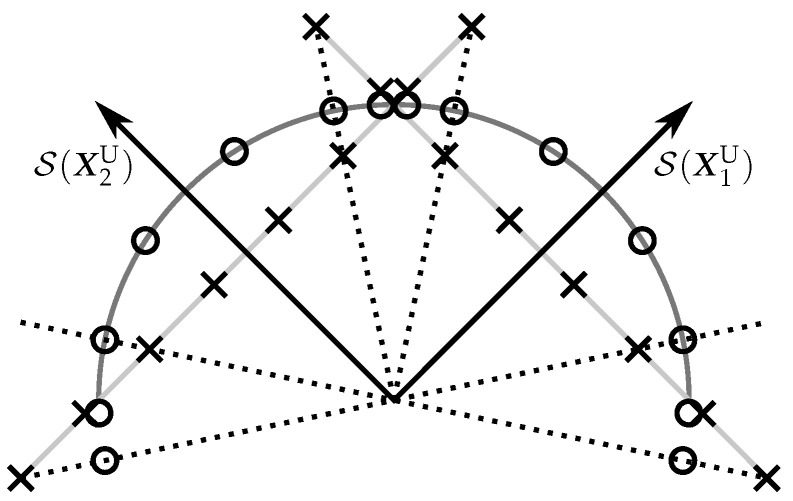
Two-dimensional graphical representation of the potential consequence of a wrong power distribution between the pilot matrix and the coherent code. Four pairs of constellation elements span the same subspaces, which are drawn with dotted lines. (element: ‘∘’ symbols, coherent code: ‘×’ symbols).

**Figure 7 sensors-22-09049-f007:**
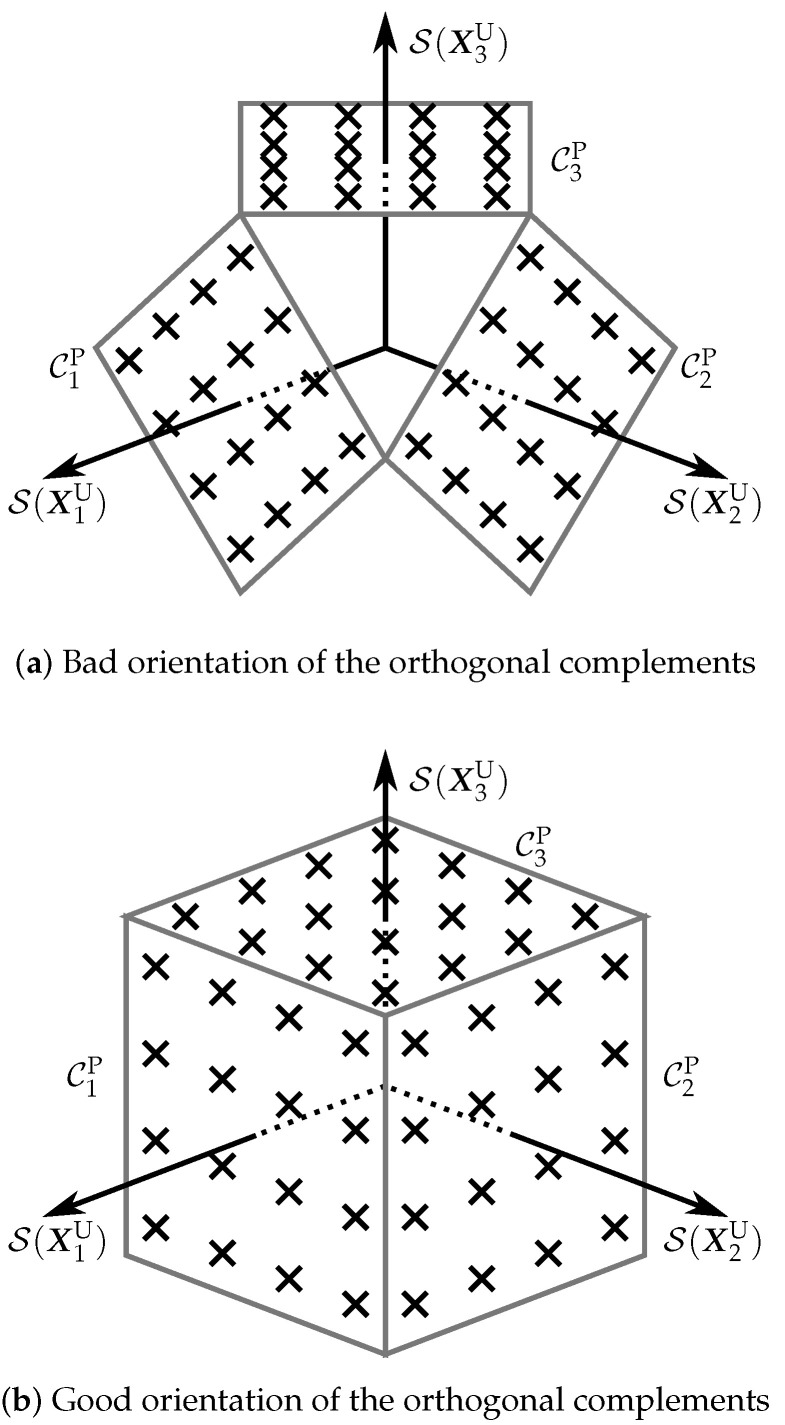
Three-dimensional graphical representation of the effect of the orthogonal complements orientation and the coherent code shape. A USTM of three elements is used to construct three PSAMs, which elements are surrounded by quadrilaterals. (coherent code: ‘×’ symbols).

**Figure 8 sensors-22-09049-f008:**
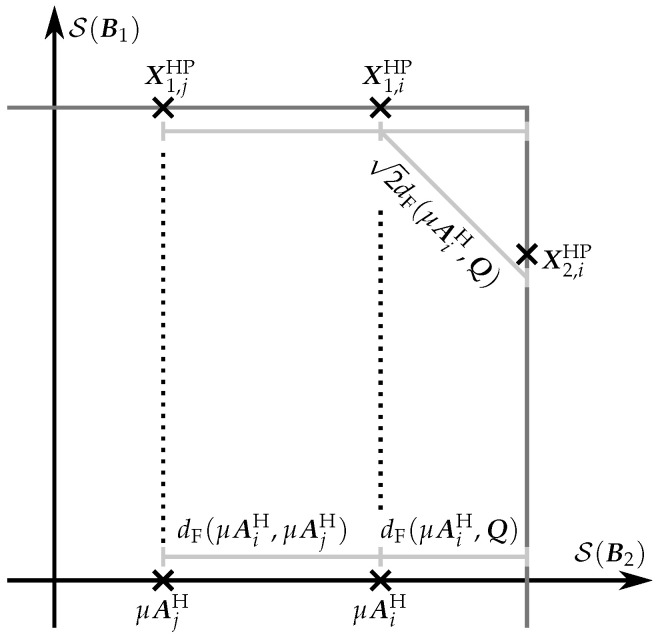
Graphical representation of the distances used in (Equation 28) to set the value of μ for $\frac{T}{M}=2$.

**Figure 9 sensors-22-09049-f009:**
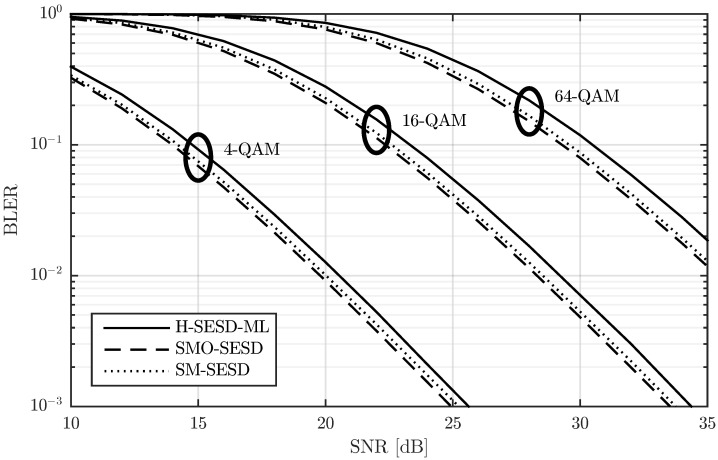
Performance comparison of three PSAMs constructed from rectangular QAM, their orthonormalization, and a constellation constructed using the hypercube method with the same QAM. All results are obtained for a channel with T=4 and M=N=2.

**Figure 10 sensors-22-09049-f010:**
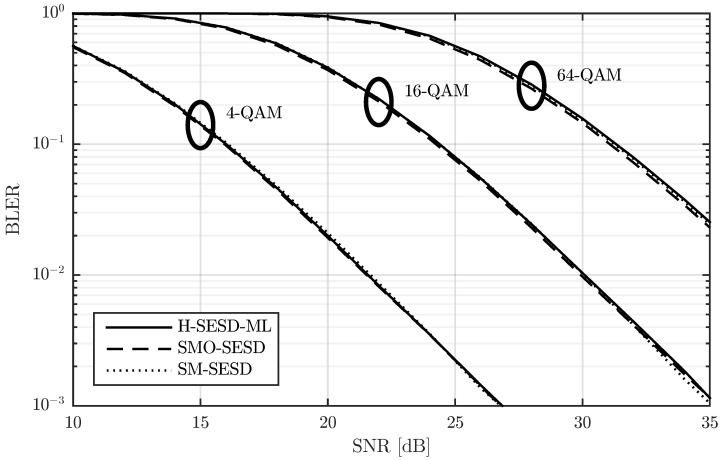
Performance comparison of three PSAMs constructed from rectangular QAM, their orthonormalization, and a constellation constructed using the hypercube method with the same QAM. All results are obtained for a channel with T=8 and M=N=2.

**Figure 11 sensors-22-09049-f011:**
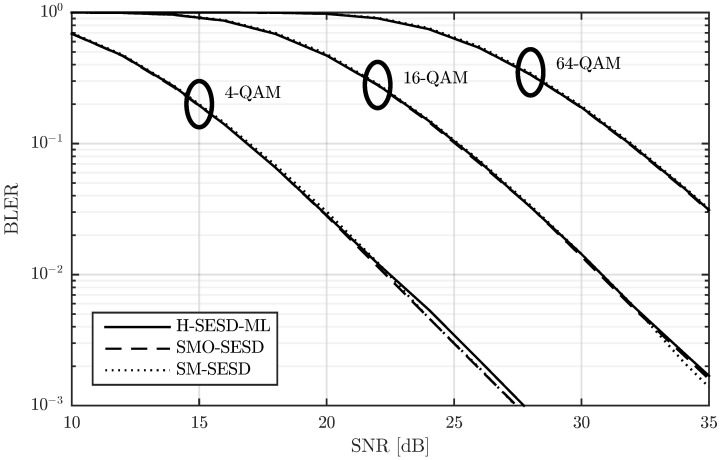
Performance comparison of three PSAMs constructed from rectangular QAM, their orthonormalization, and a constellation constructed using the hypercube method with the same QAM. All results are obtained for a channel with T=16 and M=N=2.

**Figure 12 sensors-22-09049-f012:**
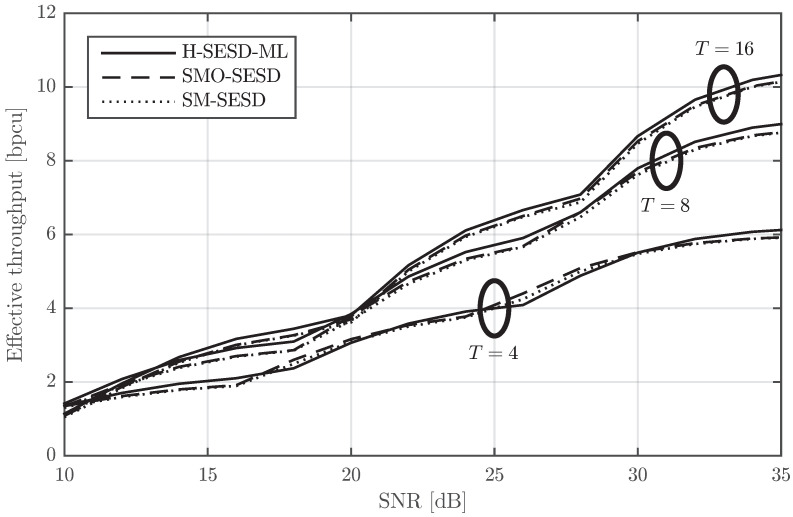
Effective throughput comparison of a PSAM constructed from rectangular QAM, its orthonormalization, and a constellation constructed using the hypercube method with the same QAM. This throughput is the maximum from that obtained with 4-QAM, 16-QAM and 64-QAM, and for M=N=2.

**Table 1 sensors-22-09049-t001:** Bits per channel use transmitted for each configuration.

T	**QAM**	**H-SESD-ML**	**SMO-SESD**	**SM-SESD**
4	4	2.25	2	2
4	16	4.25	4	4
4	64	6.25	6	6
8	4	3.25	3	3
8	16	6.25	6	6
8	64	9.25	9	9
16	4	3.6875	3.5	3.5
16	16	7.1875	7	7
16	64	10.6875	10.5	10.5

## Data Availability

Data will be provided upon request.

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
