# Peer review of "Unitary Space-Time Modulation Based on Coherent Codes"

_sensors, 2022, doi:10.3390/s22239049_

Round 1

Reviewer 1 Report

The results of the paper are interesting, I have the following concerns about the paper.

1.    The structure of the article is not proper. It has just sections of Introduction, Results, and Conclusion. So, where is related work or previous studies in this area? There should be a section of related work.

2.    References are old, related works can be expanded by including some more material using more up-to-date references.

3.    The proposed method(s) can be included in a separate section with the title of the proposed approach so that it can be distinguished from the background material.

4.    The proposed approach should be compared with similar approaches in terms of different performance parameters.

5.  Conclusion needs to be rewritten.

Author Response

I would like to thank the Reviewer for helping me to see my work from the Reader's perspective and to improve its exposition accordingly. I have modified the manuscript based on these comments, and I hope that the modifications that I have made and the responses that I have provided herein will address the Reviewer's concerns.

Point 1: The structure of the article is not proper. It has just sections of Introduction, Results, and Conclusion. So, where is related work or previous studies in this area? There should be a section of related work.

Response 1: The related works are included in the Introduction. The original manuscript has no Related Work section because the number of most related works is not very large, and they fit well in the Introduction section. I could extend the references of the paper with other not so related works in a specific section, but that would require more time than the five days provided to review the manuscript. If this Reveiwer or the Asociate Editor believe that a Related Work section is necessary for the manuscript, I would like to ask for more time to do that, in order to be able to fill the new section with high quality materials.

Point 2: References are old, related works can be expanded by including some more material using more up-to-date references.

Response 2: A similar point has been pointed out by Reviewer 2. I agree with the Reviewers that the most relevant references are old. However, this is due to the fact that the most relevant advances on the topic are also old. The original manuscript has two recent references from 2020 and 2021. There are more references like those two, but, in terms of the contents of this manuscript, they do not provide more concepts or ideas. In order to alleviate the Reviewers’ concern, I included other recent references in the revised manuscript, in particular, the references [5], [9], and [15]. These references are from 2021 and 2022. The reference [5] was included as an example of how to extend the sphere packing in the Grasmann manifold to superposition coding, cf. the last sentence in the second paragraph of the Introduction. The reference [9] was included as another example of USTM construction methods that use numerical optimization tools, cf. the first sentence in the third paragraph of the Introduction. This example focuses on the multiple access channel (MAC). The reference [15] was included as another example of mapping codes from a hypercube into the Grasmann manifold, cf. the last two sentences in the third paragraph of the Introduction.

Point 3: The proposed method(s) can be included in a separate section with the title of the proposed approach so that it can be distinguished from the background material.

Response 3: I would like to thank the Reviewer for proposing means to improve the presentation of the manuscript. The manuscript has several new construction methods, and they are separated in different sections. The organization of the original manuscript is motivated by the different methodologies of those construction methods. Although a different organization is possible, the time provided to produce the revised manuscript is not enough to do so. Due to that, I decided to maintain the original organization.

Point 4: The proposed approach should be compared with similar approaches in terms of different performance parameters.

Response 4: I believe that the Reviewer is proposing to show results of other metrics. In the original manuscript, I presented results of the Bloc Error Rate (BLER) and the effective throughput. The BLER is a very common metric in this context. Another common metric is the Bit Error Rate (BER). However, in this case, the labeling of the constellation elements is important, and the labeling optimization is out of the scope of the manuscript. It is important to note that, for a given labeling, a lower BLER implies, in general, a lower BER. In any case, new results would require new simulations, and hence, a significant amount of time to adapt the simulator, to run the simulations, to process the results, and to integrate those results in the manuscript. Due to the time provided to produce the revised manuscript, I decided to not add more results to the manuscript.

Point 5: Conclusion needs to be rewritten.

Response 5: Following the Reviewer suggestion, I have reviewed the Conclusion section. I have included some extra details to help potential Readers understand the main contributions of this manuscript.

Reviewer 2 Report

The paper analyzes the relationship between pilot symbol-assisted modulation and unitary space-time modulation. The author introduces a map that transforms any PSAM into a USTM and vice versa. Based on the relationship between PSAM and USTM, the author proposes a new USTM construction method, which ensures that the USTM has a good performance and low computational complexity.

The proposed technique is validated trough simulation and detailed results analysis.

 The references are not very new. They are no recent studies on the subject?

In abstract you write “USTMs are known to be capacity-achieving, but these constellations do not have a known structure that could simplify its decoding, and the proposed construction methods are computationally expensive”. Is not very clear if this construction method is proposed in literature or is your proposal.

Perhaps it should be stated that the channel block length T is expressed in number of symbols

What is the small square at the end of some paragraphs (e.g., line 119, 155, 309)?

The analysis around fig 10, 11, 12 suggest that the best configuration differs for different ranges of SNR and different values of T. There is a method to select the best configuration according to the values of SNR and T?

Author Response

I would like to thank the Reviewer for helping me to see my work from the Reader's perspective and to improve its exposition accordingly. I have modified the manuscript based on these comments, and I hope that the modifications that I have made and the responses that I have provided herein will address the Reviewer's concerns.

Point 1: The references are not very new. They are no recent studies on the subject?

Response 1: A similar point has been pointed out by Reviewer 1. I agree with the Reviewers that the most relevant references are old. However, this is due to the fact that the most relevant advances on the topic are also old. The original manuscript has two recent references from 2020 and 2021. There are more references like those two, but, in terms of the contents of this manuscript, they do not provide more concepts or ideas. In order to alleviate the Reviewers’ concern, I included other recent references in the revised manuscript, in particular, the references [5], [9], and [15]. These references are from 2021 and 2022. The reference [5] was included as an example of how to extend the sphere packing in the Grasmann manifold to superposition coding, cf. the last sentence in the second paragraph of the Introduction. The reference [9] was included as another example of USTM construction methods that use numerical optimization tools, cf. the first sentence in the third paragraph of the Introduction. This example focuses on the multiple access channel (MAC). The reference [15] was included as another example of mapping codes from a hypercube into the Grasmann manifold, cf. the last two sentences in the third paragraph of the Introduction.

Point 2: In abstract you write “USTMs are known to be capacity-achieving, but these constellations do not have a known structure that could simplify its decoding, and the proposed construction methods are computationally expensive”. Is not very clear if this construction method is proposed in literature or is your proposal.

Response 2: I would like to thank the Reviewer for pointing out a potential misunderstanding of this sentence. In this sentence, I was referring to most of the construction methods proposed in the literature, which focus on either maximizing the minimum distance between two constellation elements or minimizing the pairwise error probability. In this case, the constellations have a good BLER, but they lack a known structure, and the construction of the constellations is computationally expensive. In order to avoid the potential misunderstanding, this sentence has been rewritten in the revised manuscript as: “USTMs are known to be capacity-achieving. However, most of the proposed USTM construction methods in the literature are computationally expensive, and the resulting constellations do not have a known structure that could simplify their decoding.”

Point 3: Perhaps it should be stated that the channel block length T is expressed in number of symbols.

Response 3: Following the Reviewer’s suggestion, I have specifically defined T as the number of channel uses of the channel coherence interval. This definition is in the first sentence of Section 2. In particular, in the revised manuscript, this sentence is: “Consider a wireless communication system with M transmit and N receive antennas, and a block-fading channel with a coherence interval of T channel uses”. The term “channel use” is widely used in this context to refer to the smallest transmission period. Since “symbol” may have other connotations, I preferred to use the term channel use. Note that, for instance, we could say that we can transmit several symbols in parallel in a MIMO channel, or we could refer to a USTM constellation element as a symbol of the USTM constellation, although this symbol is a matrix.

Point 4: What is the small square at the end of some paragraphs (e.g., line 119, 155, 309)?

Response 4: Those squares indicate the end of a proof. They are part of the proof environment in latex, and they are automatically introduced by the latex template for MDPI papers. If I need to suppress those squares, or if the end of proofs need to be indicated differently, I kindly ask the Reviewer or the Associate Editor for instructions.

Point 5: The analysis around fig 10, 11, 12 suggest that the best configuration differs for different ranges of SNR and different values of T. There is a method to select the best configuration according to the values of SNR and T?

Response 5: As shown in the results, the best constellation and reception technique for T=8 and T=16 is H-SESD-ML. However, it is true that, for a particular T, the best QAM depends on the SNR. With respect to T=4, the best constellation and reception technique depends also on the SNR. To my knowledge, there is no analytical technique to know a priori which is the best constellation and reception technique, and which is the best QAM. This fact is not particular of the presented techniques. In fact, for systems with coherent reception techniques, we use lookup tables. For the particular case of reception techniques for QAM symbols, the lookup tables map SNR regions to the number of symbols of the QAM. These lookup tables are obtained from simulations similar to those presented in this manuscript. Indeed, Figure 12 shows the throughput vs. SNR using a proper lookup table.

Reviewer 3 Report

This manuscript is a theoretical work about the transformation of PSAM and USTM. A system model is introduced for the PSAM matrices and a map for the transform is illustrated. The performance of this method has been evaluated compared with the traditional SM-SESD and SMO-SESD and turns out to be better in terms of BLER. The overall illustration is clear and I don’t have further questions. I would recommend accepting this manuscript as is. 

Author Response

I would like to thank the Reviewer for his/her comment about the paper. No issues were highlighted by the Reviewer and, as a consequence, I have not modified the paper with respect to this Reviewer’s concerns.